# Impact Analysis of Cyber Attacks against Energy Communities in Distribution Grids

Afroz Mokarim *, Giovanni Battista Gaggero [ID] and Mario Marchese [ID]

Department of Electrical, Electronic and Telecommunications Engineering, and Naval Architecture—DITEN, University of Genoa, Via Opera Pia 11A, 16145 Genoa, Italy; giovanni.gaggero@unige.it (G.B.G.); mario.marchese@unige.it (M.M.)
* Correspondence: afroz.mokarim@edu.unige.it

**Abstract:** With the advancement of regulations regarding the reduction in carbon emissions, renewable energy communities have come into the picture. However, many implications come with the installation of these communities from a cybersecurity point of view. The software platforms responsible for managing and controlling them handle a lot of crucial information, and therefore, tampering with these data can lead to several impacts on the operation of these communities and, in turn, the power grid as well. This paper examines the plausible impacts that can be caused by altering certain parameters of the system that make it a potential cyber attack target. The analysis is done by observing how the grid responds to these manipulations for both low-voltage as well as medium-voltage systems. These systems are designed along with integrated energy communities and are implemented in MATLAB/Simulink R2022b software. The observations are made by plotting the grid voltage and power profiles in normal as well as attacked conditions.

**Keywords:** renewable energy communities; cyber attack; European low-voltage test feeder; power manipulation; load demand

## 1. Introduction

The Renewable Energy Directive (2018/2001/EU), commonly referred to as REDII, presents a legal framework aimed at advancing renewable energy sources (RESs) and involving citizens in the energy transition. This is achieved through the introduction of two novel mechanisms: collective self-consumption schemes (CSCs) and renewable energy communities (RECs). These instruments are built upon the principles of local energy production from RESs and the sharing of energy among end users. CSCs involve users within the same building, while RECs encompass those in the proximity of the production facility. The implementation of REDII in the national legislation of European Union member states has led to the emergence of RECs. This, in turn, has generated demand in the market for software platforms capable of streamlining the administrative, financial, and technical management of these communities. To operate effectively, these platforms handle sensitive information, including real-time electricity consumption data from REC members. They also possess the capability to regulate components such as heat pumps and battery energy storage systems (BESSs). The potential threat of a cyber attack looms large and poses a dual risk. On the one hand, a cyber attack could disrupt a building's heating system or compromise the privacy of occupants by analyzing consumption patterns. On the other hand, if a significant number of generators were manipulated maliciously, it could pose a broader risk to the entire distribution grid. The main contribution of this paper is to analyze the risk associated with cyber attacks on smart energy communities through the quantitative analysis of the impact on the distribution grid. Results can be used during a risk assessment phase of new energy communities, which avoids the need to build custom simulations, thanks to the generalizable use cases of this work. The main hypothesis is that the central platform contains common vulnerabilities for web servers that could be

exploited by attackers; under this hypothesis, results show how the impact of an attack is not limited to economic damage but can also lead to severe issues for the distribution grid, up to the interruption of services. The results can be used by distribution system operators and REC designers to minimize the risk during the development of the system; in particular, results show how, under certain conditions, the impact on the grid could be critical, therefore requiring more stringent security countermeasures. The paper is structured as follows. Section 2 analyzes the related works regarding the impact on the grid of cyber attacks and cybersecurity issues in energy communities. Section 3 analyzes the REC communication architectures, identifies the attack surface, and defines the attack model of the present work. Section 4 presents the use case scenarios that are modeled in order to quantitatively measure the impact of the attacks. Section 6 discusses the results, analyzes the implications for distribution system operators (DSOs) and REC designers, and provides some possible research directions. Finally, in Section 7, conclusions are drawn.

## 2. Related Works

The risk for the availability of power systems related to cyber attacks has been increasingly studied in the last few years. Extensive initiatives have been undertaken within the power sector to address cybersecurity concerns related to smart grids. One significant effort is the work done by the National Electric Sector Cybersecurity Organization Resource (NESCOR) [1]. NESCOR has outlined the architecture and established cybersecurity requirements for distributed energy resources (DERs) based on the framework proposed in "National Institute of Standards and Technology Interagency Report 7628" [2]. The authors in [3] studied the consequences of a denial-of-service (DoS) attack that renders the active power dispatch message inaccessible from the energy management system (EMS) to the energy storage system (ESS). This attack becomes particularly detrimental when the power output of intermittent distributed energy resources (DERs) such as wind turbines or photovoltaic (PV) systems abruptly declines due to changes in weather conditions, resulting in power imbalances and frequency deviations.

Reference [4] analyzed the consequences of an electric vehicle botnet on both an IEEE 33-bus distribution network and an IEEE 39-bus transmission model. Reference [5] manipulated actual load demand within an IEEE nine-bus system to demonstrate the adverse effects of similar attacks. In a comprehensive analysis focused on the security of EV charging infrastructure [6], the authors explored the vulnerability of the power system to attacks targeting EVs. The study simulated impacts on distribution levels, such as system oscillations, with the aim of quantifying the risks posed to distributed energy resource equipment.

Furthermore, the susceptibility of charging infrastructure is contingent on the accessibility of electric vehicle charging stations (EVCSs). Reference [7] investigated the repercussions of Internet of Things (IoT)-enabled cyber attacks on transmission and distribution grids. Reference [8] delved into the implications of attacks on power system frequency by conducting a transient stability analysis by inducing scenarios that manipulated the frequency to drop below 59.5 Hz and rise above 61.5 Hz—critical operating regions. Such attacks could lead to events like load shedding and outages, especially if the adversary coordinates attacks on a large number of EV charging points, as discussed in [9]. Reference [10] assessed the impact in terms of economic losses, reliability indices, voltage stability, and power losses within a distribution network simulated on an IEEE 33-bus system Reference [11], similarly to this work, analyzed the impact of a cyber attack on a low-voltage distribution system, but the authors took into account specifically the use case of electric vehicle charging systems; while electric vehicles can be part of RECs, the profile of controllable power over a specific area for energy communities drastically changes, and therefore a dedicated analysis is necessary. Ref. [12] discussed a scenario wherein distributed energy resources (DERs) are compromised in a cyber attack, resulting in the manipulation of their power output. This could lead to continuous oscillations or even destabilize the system. However, the authors did not deeply explore how manipulating

DERs in specific locations could amplify the impact of such attacks. Similarly, ref. [13] assessed the ramifications of controlling a multitude of DERs, albeit with a focus on storage solutions. Ref. [14] demonstrated potential violations of voltage limits via cyber attacks on DERs within the context of the CIGRE medium-voltage benchmark grid. Despite these insights, there remains a scarcity of research comprising simulations conducted to account for the unique features and limitations of utilizing a single technology to manage a large array of DERs.

Energy communities are a promising field of research. Most works focus on the economic and technical advantages of the implementation of this technology. Since the technology is very young, the cybersecurity implications have not yet been properly investigated. The first work that specifically took into account the cybersecurity issues in smart energy communities was [15]; in this paper, the authors identify the attack surface and identify three main attack vectors: attacks against the platform, attacks against the smart gateways, and attacks on the communication stack. Still, a quantitative analysis of the impact of such an attack has not been provided.

## 3. Cybersecurity Issues in Renewable Energy Communities

### 3.1. Platforms for Managing RECs

Members of a renewable energy community (REC) utilize the existing distribution grid to share the energy they generate and adhere to proximity constraints between generation and consumption. The specific interpretation of the term "proximity" is delegated to national regulations governing RECs. For instance, in countries like France [16] and Spain [17], energy sharing is permissible only if members are linked downstream of the same secondary substation. In contrast, in Italy [18], it is allowed if members are under the same primary substation. The communication network for these communities, as well as the power infrastructure, does not rely on a dedicated infrastructure but extensively utilizes the public internet. An overall scheme is shown in Figure 1.

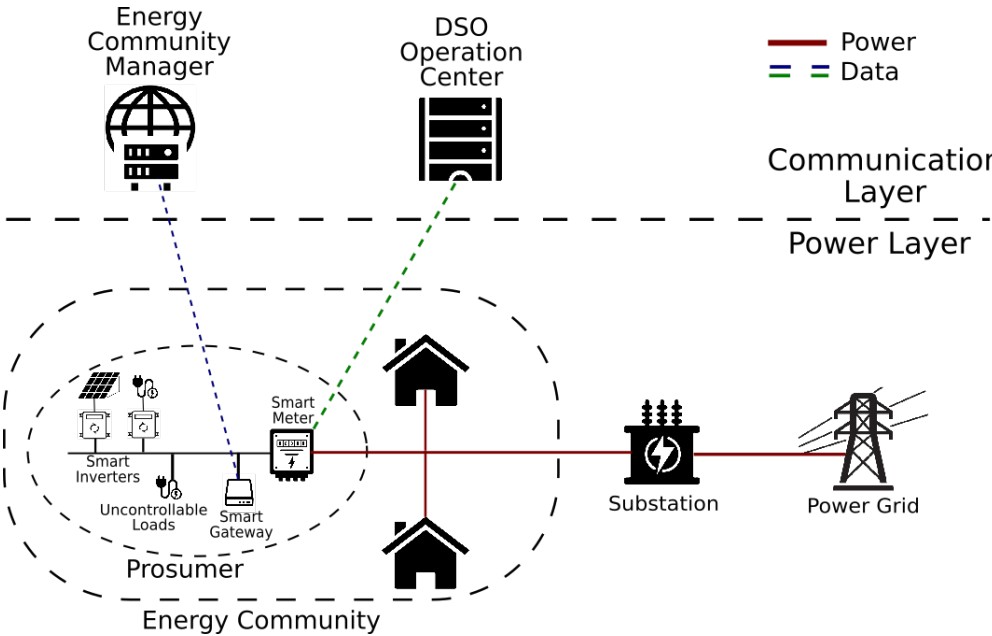

**Figure 1.** Overall scheme of an energy community.

Generators and flexible loads within the renewable energy community need to communicate to effectively coordinate their actions and achieve economic or technical objectives. Additionally, there is a demand for software that can facilitate coordination among community members. Consequently, the establishment of RECs has given rise to the development of platforms designed for managing energy communities, often in the form of web applications tailored for RECs.

In this context, the imperative need for the development of robust software platforms to manage energy communities is evident. Reference [19] proposes a web-based platform designed for the administration of energy communities that addresses aspects such as energy tariffs, aggregation of end-users, price elasticity, and load response. Emphasizing the scalability of software solutions, reference [20] underscores the significance of maximizing the value derived from distributed green assets, including photovoltaic systems (PVs), batteries, electric vehicles, and electric heating and cooling. Notably, there are various commercial solutions available for such platforms. One such product is the Regalgrid platform [21]: as described by the manufacturer, it enables sophisticated management of energy resources by interfacing with various device types via the SNOCU controller [22]. This facilitates a comprehensive view of one's energy profile and optimizes consumption management. The SNOCU controller exemplifies an advanced gateway that provides remote oversight of generators and is often referred to as a "smart gateway" by its producers. Another innovation is the ROSE energy platform [23]: a cloud-based service for establishing, simulating, and overseeing energy communities. This service includes an energy management system (EMS) module and a mobile application designed to engage community members. Additionally, ER-LIBRA CE [24] represents a cloud-based solution dedicated to the administration of energy communities and self-consumption groups and incorporates a control module for storage system management. We notice how most of the currently available commercial products are based on a centralized platform, similar to the reference architecture shown in Figure 1.

Renewable energy community (REC) platforms serve multiple functions, including a dashboard for administration, real-time monitoring of the REC, real-time monitoring of the members' treasury, and control of DERS through an energy management system (EMS). In particular, the malicious exploitation of the last function is the core of this paper.

*3.2. REC Communication Architectures*

Usually, in a power distribution system, there is no or little communication between the consumers/prosumers and the DSO. The only relatively common communication infrastructures in Europe for this purpose are advanced metering infrastructures (AMIs). In this case, the smart meter exchanges data with the DSO center, usually for billing purposes. In this scenario, if there is a need to control generators and/or loads, the users need to install additional devices. For web-platform-controlled energy communities, users need to install an additional device that we call a smart gateway: the smart gateway has the aim of managing from one side the internal communication, for example sending power set points to the inverters, and from the other side the communication with the energy community manager platform. The smart gateway, therefore, is the crucial device that, from the network's perspective, act as a network gateway for converting web protocols into local communication protocols such as Modbus, 802.11, Bluetooth, etc.

Therefore, in a smart energy community controlled by a web platform, we can identify three main communication channels, as shown in Figure 1:

- Local communication: this is the communication channel used by the apparatus of the prosumer to exchange data and commands, such as the communication between the smart gateway and the smart inverters.
- Smart meter—DSO: in many countries, the DSO owns an AMI for billing, monitoring, and control purposes; strictly speaking, this is not part of the energy community and is not considered an attack model in our paper.
- Smart gateway—energy community manager: this usually makes use of the public communication network or the mobile network and of common web protocols.

This means that both the smart gateway and the platform can be reached by the public network and are susceptible to attacks. Also, the power distribution system can be prone to attacks on the communication stack.

### 3.3. Attack Model

In this scenario, the hypothesis is that if a cyber attack against the central platform succeeds, the attackers can control all the apparatus that the platform can control in normal conditions but with the aim of disrupting the operation of the distribution grid. Therefore, the platform represents a single point of failure for the whole infrastructure wherein many common attacks on servers and web applications could play a role in the attack model of this work.

Usually, the platforms can only control the set points of active and, in some cases, reactive powers of all the generators and controllable loads. Under normal conditions, the power installed is supposed to work with a certain utilization factor and load factor or, alternatively, to follow specific logic with respect to the condition of the distribution grid. For example, not all the loads are supposed to absorb the maximum active power in the same instant of time, and the storage systems are not supposed to inject the maximum active power while the photovoltaic systems are generating energy. Nevertheless, since the platform has the capability of modifying the set point, in case of attack, the whole system can behave in a way that is dangerous for the distribution grid. Therefore, the attacks in all the following use cases are simulated using manipulation of active and reactive power sources of the system in consideration.

The attack model scheme is shown in Figure 2. As shown, the scheme depicts the REC's connections, including the internet services and the power grid. The connection can make use of many protocols like IEEE 1815 [25] and IEEE 2030.5 [26] to name a few. The data being communicated by the REC via the smart gateway to the management system are measured by the sensors and smart meters and include the active and reactive power, frequency, voltage, current, and power factor. In turn, the management system handles the inputs and outputs of quantities by controlling the REC infrastructure. The attacker can thus manipulate the data being used to control the REC devices in two main ways: from one side, he can exploit vulnerabilities of the EMS server, such as through cross-site scripting [27], SQL injection, and cross-site request forgery [28]. Also, he can target the smart gateways that, since they are usually IoT devices built on low-computational-power devices, may suffer from severe vulnerabilities. In both cases, the impact is the manipulation of the load parameters and active and reactive power references and switching the operation modes of the energy-producing devices.

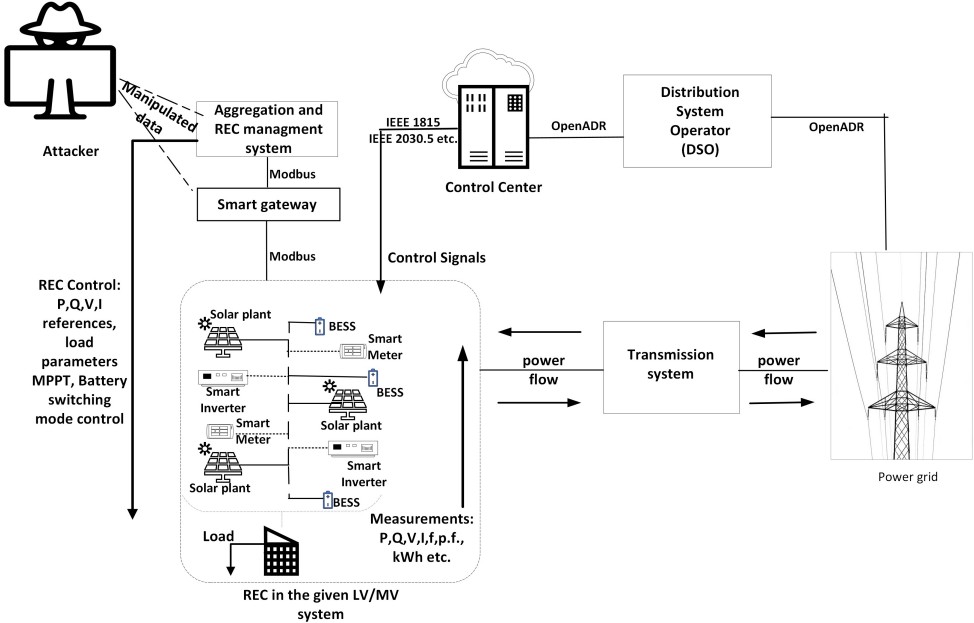

**Figure 2.** Attack model scheme.

## 4. Use Case Scenarios

Initially, the power constraints and boundaries on RECs were defined by "Article 42-bis of Law No. 8" [29], which allowed them to be only located on low-voltage grids and restricted them to generating and consuming not more than 200 kW of plant power. This was later updated by "Legislative Decree No. 199" [18], which eased the above power constraints from 200 kW to 1 MW, allowing a larger number of prosumers to participate. However, taking into account general examples with low-voltage distribution systems such as residential buildings [30], a deployment of 200 kW is considered to be optimal. We take into account the two main cases that, according to national laws that establish economic incentives for RECs, could be designed by engineers: a maximum of 200 kW under a single low-voltage (LV) substation and a maximum of 1 MW under a single primary medium-voltage (MV) substation.

### 4.1. Low Voltage (LV)

An ideal case for representing a low-voltage distribution system is the classic IEEE European low-voltage test feeder (ELVTF) [31], as shown in Figure 3. It represents a radial distribution network with 906 nodes and a phase-to-phase voltage level set at 416 V, operates at a base frequency of 50 Hz, and incorporates 55 load buses in its configuration. As indicated in the figure, the feeder is divided into three areas. According to the number and location of the loads, Area 1 and Area 3 are assumed to have residential prosumers with total installations of 50 kW each, comprising a combination of PV plants and battery energy storage systems (BESSs). Similarly, Area 2 is assumed to be composed of prosumers with 100 KW of maximum power installations. The maximum capacity of the PVs as well as the BESSs is 200 kW each, which means that whenever the PV is not in operation, the BESS is supposed to supply the required power. However, only one of them is supposed to be under operation at a certain time. Hence, overall, the total installed power of this energy community does not exceed 200 kW. The basic scheme of this energy community is given in Figure 4. It is also worth mentioning that the maximum load subjected to the system from all three areas is approximately 234 kW.

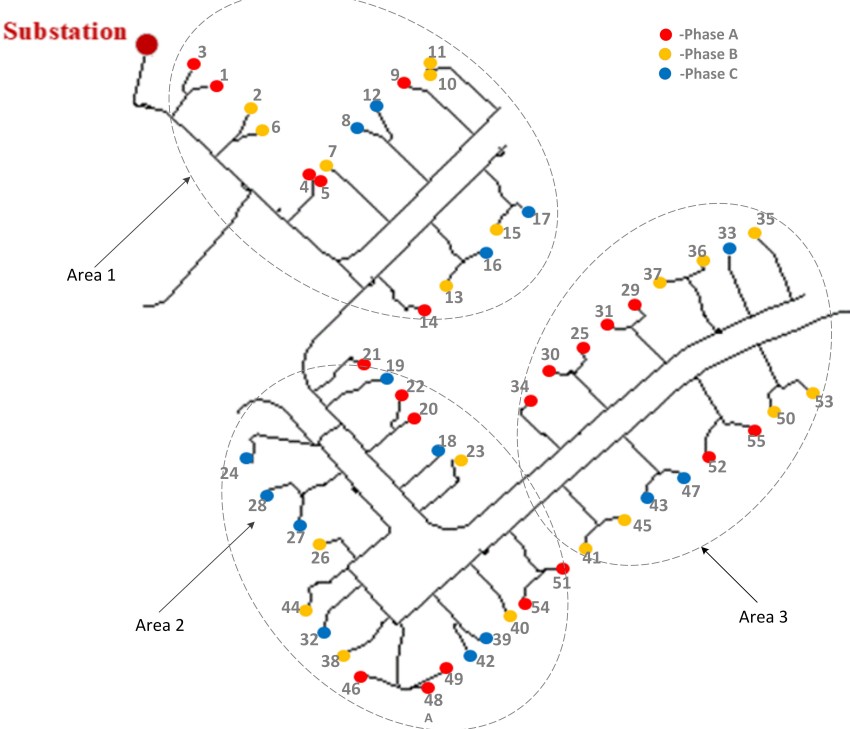

**Figure 3.** IEEE European low-voltage test feeder (ELVTF).

When the adversary wants to attack the given system, obviously he has to cause power manipulations in such a way that either the system goes into a shortage of power or a surplus of power. This is because either of them can be dangerous to the grid voltage profile if they occur at certain times of the day according to the load demand trend.

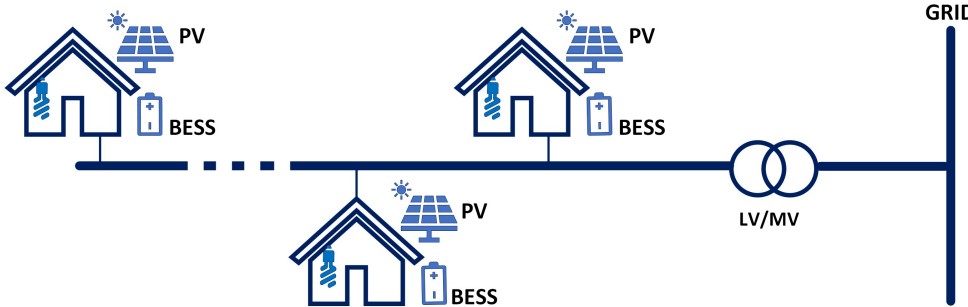

**Figure 4.** Energy community scheme implemented in use case scenarios.

For example, a generic daily load demand trend of Italy is shown in Figure 5. The figure shows the trend of instant active power demand in the whole country on a generic day in January (which is the coldest month of the year) taken from the website of the Italian transmission system operator Terna s.p.a. [32]; it can be noticed that the load demand rises from 8 a.m. and reaches its peak around 11 a.m. It then starts to show a decline from 12 p.m. onward and again reaches a peak in the evening around 6 p.m. The load demand is lower around 11 p.m. and keeps on declining. Therefore, if the adversary is able to plan the attacks according to the load demand, they could have a worse impact on the grid. Even at high loads, either the PV plant or the BESS can supply the grid with sufficient energy to not cause any fluctuations in the voltages or currents. However, a problem can arise when both of them are switched on or off simultaneously with different loading conditions.

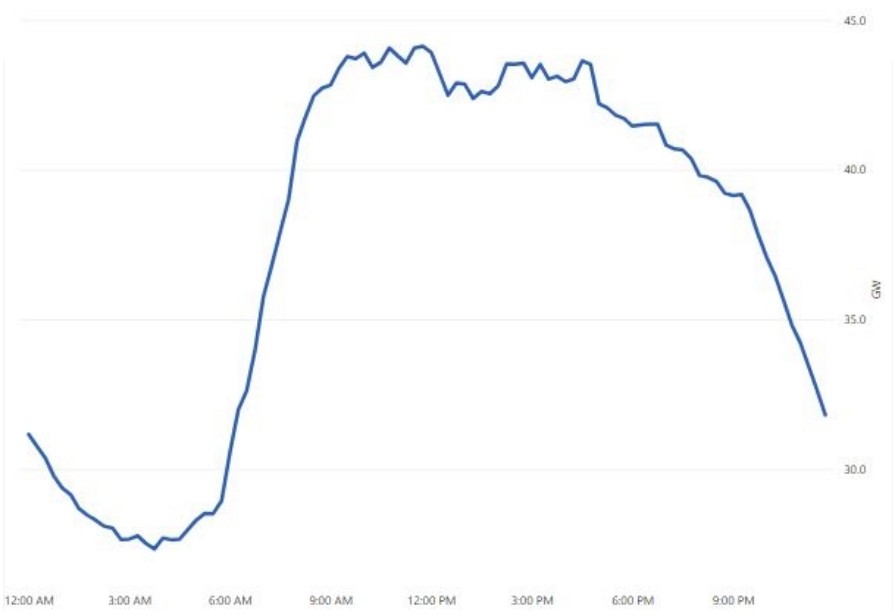

**Figure 5.** General load demand trend in Italy.

Within the same use case, the power manipulation attack is executed in three different scenarios, as shown in Table 1. The power manipulation attack can lead to higher absorption of power from the grid to the loads (in the case of controllable loads such as heat ventilation and air conditioning systems) or to higher production (injection) of power to the grid (for example, in case of battery energy storage systems). We simulated three different scenarios

for which it is possible to control a different magnitude of power across the three areas considering a realistic deployment of resources across the distribution grid. For example, Area 2 is subjected to higher magnitudes of power manipulation due to the centralized position of loads.

**Table 1.** Attack scenarios for LV system.

|  | Case 1 (Injection) | Case 2 (Absorption) | Case 3 (Injection) |
|---|---|---|---|
| Area 1 | 50 kW | 50 kW | 100 kW |
| Area 2 | 100 kW | 100 kW | 200 kW |
| Area 3 | 50 kW | 50 kW | 100 kW |
| Total power | 200 kW | 200 kW | 400 kW |

As shown in the table, the power is either injected or consumed according to different situations.

### 4.2. MV

For representing the MV systems, a IEEE 69-bus system is chosen that works at a nominal voltage of 12.67 kV and with a base power of 10 MVA. The total active load on this system is 3800 kW, and the total reactive load is 2690 kVAr. The system has 69 nodes connected by 73 branches. For this study, the peak power limit for the entire REC is 1 MW according to "Legislative Decree No. 199" as mentioned previously in this section. Also, being an MV system, it presents a much larger area, and hence, the REC is divided into sub-communities placed at different arbitrary points in this system. The single-line diagram of the 69-bus system is shown in Figure 6. The diagram can be seen with the sub-RECs placed between various buses. The attack scenarios for the MV system are presented in Table 2. Four cases of power manipulation are considered for the study of MV systems; the first two deal with the injection and absorption of active power, whereas the next two deal with the injection of inductive (".ind" in Table 2) and capacitive (".cap" in Table 2) reactive power. The study of reactive power at the MV stage becomes relevant, as high magnitudes of reactive power can also alter the grid profile.

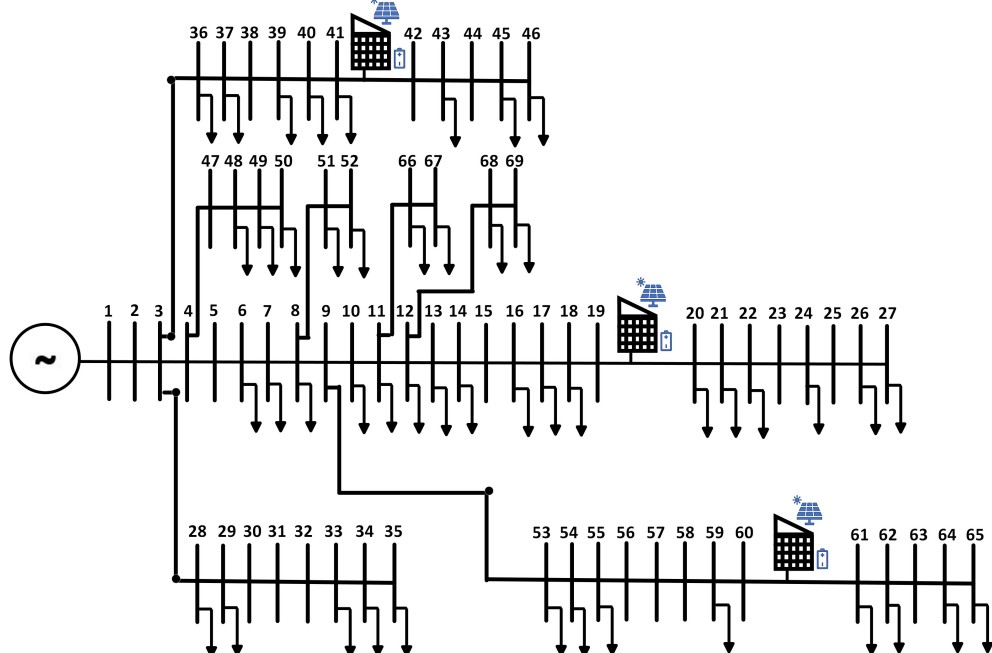

**Figure 6.** Single-line diagram of IEEE 69-bus system integrated with REC.

**Table 2.** Attack scenarios for MV system.

| | Case 1 (Injection) | Case 2 (Absorption) | Case 3 (Injection) | Case 4 (Injection) |
|---|---|---|---|---|
| Bus 41 | 800 kW | 800 kW | 400 kVar (ind.) | 400 kVar (cap.) |
| Bus 19 | 800 kW | 800 kW | 400 kVar (ind.) | 400 kVar (cap.) |
| Bus 60 | 400 kW | 400 kW | 200 kVar (ind.) | 400 kVar (cap.) |
| Total power | 2 MW | 2 MW | 1 MVar (ind.) | 1 MVar (cap.) |

## 5. Results

This section provides all the observations when the two given systems are manipulated according to the above tabular data.

### 5.1. LV

In the IEEE ELVTF system, the impact of these attacks is analyzed by studying the voltage profile of the grid. Hence, after the attacks, the impacts on the voltage profiles of load 1, load 34, and load 50 are studied. The system is implemented on MATLAB/Simulink R2022b, and all the cases are executed with the resultant voltage plots shown in subsequent figures.

#### 5.1.1. Case 1: PV Supply on, Battery Switched on, Increased Load Demand

In this case, a general sunny morning is considered, during which the PV is supplying at its full capacity of 200 kW, the BESS is, therefore, in off mode, and the load demand is also high according to the trend shown in Figure 5. However, if the adversary causes the BESS to turn on and start supplying power during this period, it would mean a surplus of 200 kW injected into the system. This can cause the grid voltage to rise up to as high as 1.12 pu, as shown in Figure 7. The plot in this figure shows the voltages taken at load 1, load 34, and load 50 with respect to Figure 3 between 8 a.m. and 11 a.m. The measure taken near load 1 corresponds to the voltage at the substation, while loads 34 and 50, being located at two different distances from the substation, are representative of the voltage profile across the line. Load 34 seems to undergo a worse impact owing to its central location and the magnitude of the power manipulation, as given in Table 1.

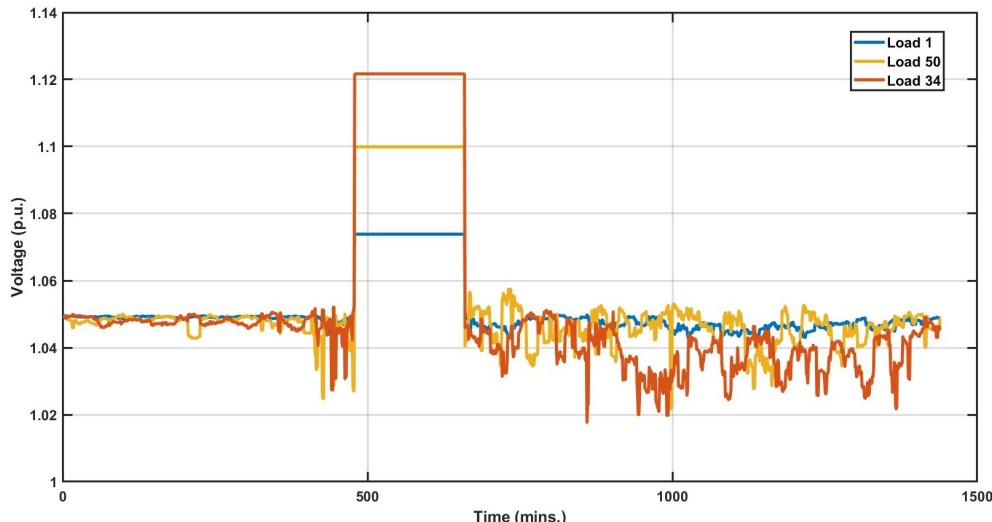

**Figure 7.** Voltage profile for an injection of 200 kW active power.

#### 5.1.2. Case 2: PV Supply Off, Battery Charging, Increased Load Demand

This case takes into account the fact that on certain days, the community might not have a PV supply because of negligible sunlight, and the battery is not supplying either. However, during this time, the adversary can put the BESS into charging mode, causing it to

become an additional load. Also, choosing a peak load time like morning for manipulating the load is a further risk. This might cause the system to be overloaded at approximately 400 kW. The resultant voltages for loads 1, 34, and 50 is plotted and is shown in Figure 8. The figure shows a maximum voltage drop down to 0.87 pu between 8 and 11 a.m., which is outside the safety limit of 10% as provided in the grid codes. However, for load 1, which is near the substation, the impact of the attack is manageable.

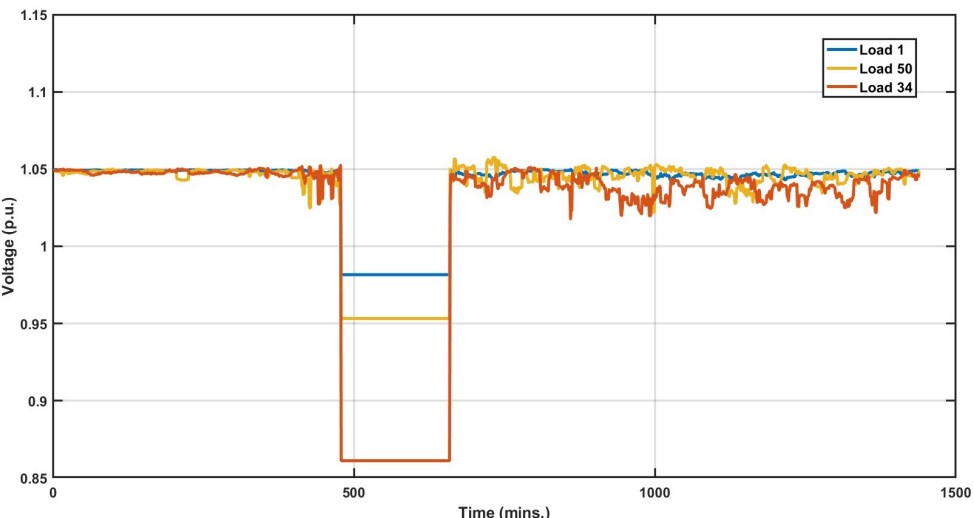

**Figure 8.** Voltage profile for an absorption of 200 kW active power.

### 5.1.3. Case 3: PV Supply on, Battery Supply on, Decreased Load Demand

This case can happen in the early morning when the PV is supplying and load demand is lower comparatively. During these hours, the adversary can control the battery again to start supplying power alongside the PV. This can again cause a surplus of power to be injected into the system like in Case 1 but at a larger magnitude. The system was subjected to these power conditions and results were observed between 5:30 a.m. and 7 a.m., which can be seen in Figure 9. The figure shows that this attack has a much larger impact on the grid and causes the voltage to reach up to approximately 1.18 p.u. The substation might also get disturbed, as the voltage rise reaches around 1.1 p.u. there.

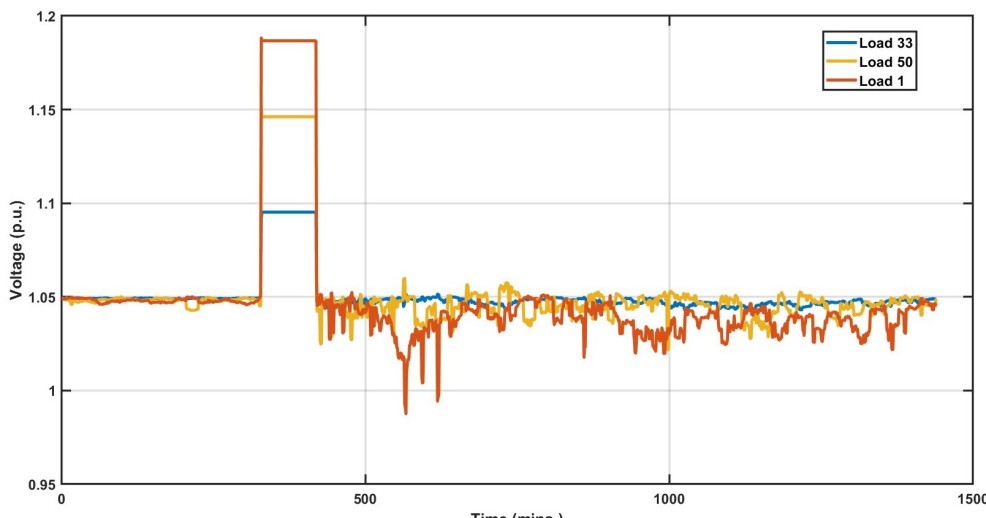

**Figure 9.** Voltage profile for an injection of 400 kW active power.

*5.2. MV*

To analyze the consequences of the attacks on the MV system, quantities such as voltage, current, and active and reactive powers have been plotted for bus 13, which is somewhat at the central part of the grid, and also for bus 1 which is near the substation. The locations of these buses are important from an observational point of view. Unlike the LV system, the results for this system are plotted for 10 s to show the immediate effects of the attacks for the following cases.

5.2.1. Case 1: PV Supply on, Battery Switched on

In this case, the PV is supplying active power of 1 MW, and the battery is therefore in off mode. The attacker can therefore take advantage of controlling the battery and turn it on to supply power at its full capacity, which is, again, 1 MW. This causes a surplus of power and results in the grid voltages and currents rising more than their nominal values. The first graph in Figure 10 shows that the voltage dangerously rises to approximately 1.8 p.u. at bus 13, which is at the central location. The effects near the substation are lower at around 1.3 p.u., but they still amount to an increase of 30% from the nominal value. The voltage and current relationship depicted in the second graph shows that they are in phase with each other. The third graph shows the total injected active power, which is approximately 2 MW, and zero injected reactive power at bus 13.

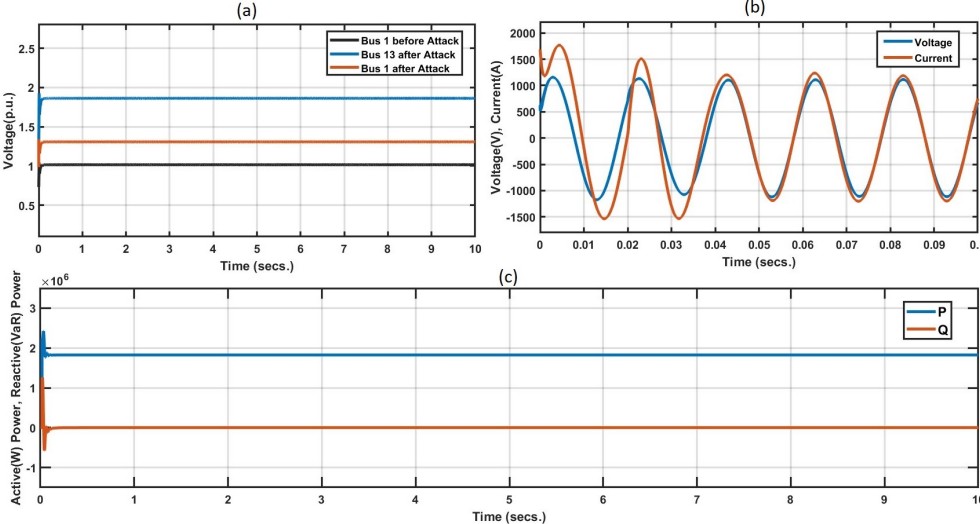

**Figure 10.** Grid parameters for 2 MW active power injection: (**a**) p.u. voltage profiles of grid buses, (**b**) voltage and current waveforms at time of attack, and (**c**) total active and reactive power measured at bus 13.

5.2.2. Case 2: PV Supply Off, Battery Charging

This case is concerned with times when the PV is not in operation and the required energy is supplied by the battery. Here again, the battery can be controlled and put into charging mode. This makes it behave as an additional load on the system and causes disturbances in the voltage and current profiles as well.

The first plot in Figure 11 shows that the p.u. voltage drops considerably down to 0.75 p.u. at bus 13. The drop near the substation is around 0.88 p.u.: amounting to a decrease of 12% from the nominal value. The voltage and current relationship is depicted in the second graph, which shows how the given attack puts the grid current out of phase with the grid voltage. The third graph shows the total absorbed active power, which is 2 MW (shown in negative), and zero reactive power at bus 13.

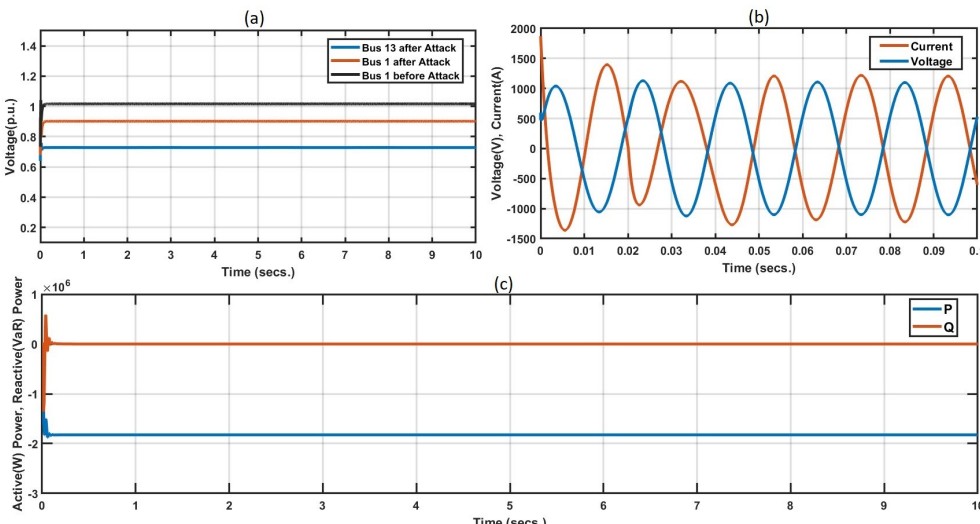

**Figure 11.** Grid parameters for 2 MW active power absorption: (**a**) p.u. voltage profiles of grid buses, (**b**) voltage and current waveforms at time of attack, and (**c**) total active and reactive power measured at bus 13.

5.2.3. Case 3: Injection of Inductive Reactive Power

In this case, the adversary can cause manipulation of the reactive power input to the system. The battery as well as PV inverters can be controlled to supply a large amount of inductive or capacitive reactive power to the grid in addition to the active power. The plots in Figure 12 show the effects of injecting an inductive reactive power of 1 MVAr.

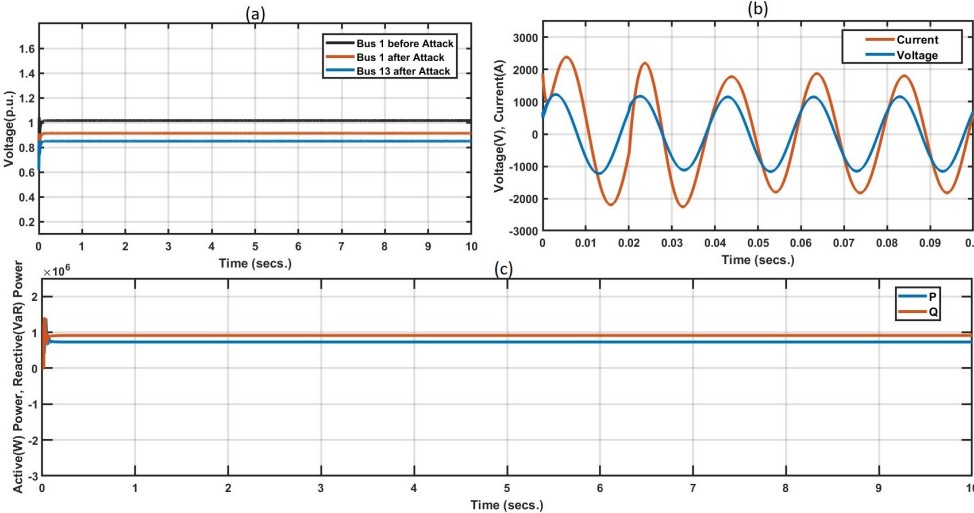

**Figure 12.** Grid parameters for 1 MVAr inductive reactive power injection: (**a**) p.u. voltage profiles of grid buses, (**b**) voltage and current waveforms at time of attack, and (**c**) total active and reactive power measured at bus 13.

The grid voltage drops due to the inductive nature of the reactive power, which makes the voltage drop down to 0.8 p.u. at bus 13. The drop near the substation is around 0.9 p.u., giving a decrease of 10% from the nominal value. The drops have smaller magnitudes because of the injection of some active power as well. The voltage and current relationship depicted in the second plot shows the grid current lagging the grid voltage, which confirms the inductive nature of power injection. The third graph shows the total injected active power at 800 kW and inductive reactive power of 1 MVAr at bus 13.

### 5.2.4. Case 4: PV Supplying Reactive Power, Battery Charging

Similar to the previous case, the attacker can manipulate the PV system to inject capacitive reactive power into the grid. The first plot in Figure 13 shows that the voltage rises to 1.2 p.u. at bus 13. The rise near the substation is approximately 1.12 p.u., amounting to an increase of 12% from the nominal value. The voltage and current relationship is depicted in the second graph, which shows the grid current leading to the grid voltage. The third plot shows the reactive power as a negative value at bus 13. Also, the active power is absorbed (shown as a negative value in the plot) as the battery is put in a charging state. This case shows double manipulation of both active as well as reactive powers, rendering the entire system very vulnerable.

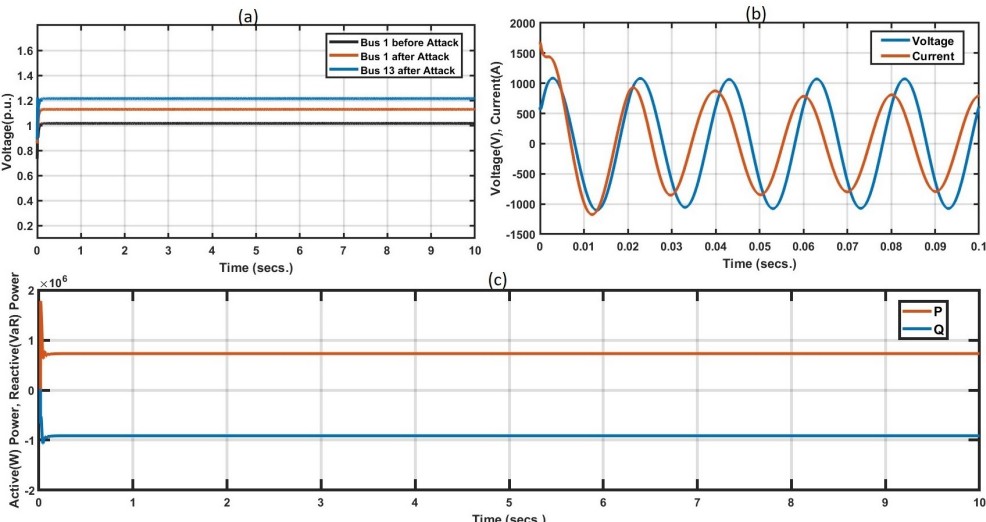

**Figure 13.** Grid parameters for 1 MVAr capacitive reactive power injection: (**a**) p.u. voltage profiles of grid buses, (**b**) voltage and current waveforms at time of attack, and (**c**) total active and reactive power measured at bus 13.

## 6. Discussion

The effects of power manipulation attacks can be seen in both the LV and MV systems. For the LV system, the voltage profile is plotted for three cases, which all involve different scenarios of power manipulation. The IEEE ELVTF is taken for this study and is subjected to a surplus of 200 kW and 400 kW as well as a deficit of 400 kW. All the scenarios have been plotted, and most of them either show a voltage drop or a rise of more than 10% of the nominal value. This violates the constraints of the grid and puts the grid at risk. In this scenario, protection equipment should likely be configured to protect the grid from sudden shocks, but this can lead to a major interruption of services to the consumers anyway. The timing of these attacks makes them even more dangerous if the general load demand trend is taken into account.

For the MV system, which has a higher magnitude of power installed, the problem of manipulating reactive power is also considered. Four scenarios have been taken wherein the IEEE 69-bus system taken for this study is subjected to not only active power injection and absorption but also surpluses of inductive as well as capacitive reactive powers. The attached plots show voltage, current, and active and reactive power behaviors immediately after the introduction of attacks. The sudden surplus or deficit of more than the allotted power can make the RECs responsible for the grid's decreased performance.

For these reasons, an attack on an energy community in a low-voltage grid can have a much more severe impact than in a medium-voltage grid. In particular, while in an MV grid, the attack impacts the grid primarily at the local level, in the LV grid, this could even cause disconnection of the whole substation. These considerations should be taken into account by distribution system operators while evaluating the cyber risk associated

with these systems. While from one the side, several services can be used to monitor known vulnerabilities on the employed devices, such as the "Common Vulnerabilities and Exposures" (CVEs) provided on the "MITRE CVE List", it is practically unfeasible to dispose of an updated asset inventory for the huge variety and quantity of devices managed by a single utility. Moreover, cybersecurity in the operational technology environment must take into account attack response strategies that guarantee the availability of the main power system. This risk has to be taken into account, especially since little automation is generally implemented in low-voltage grids. DSOs can generally decide to disconnect electrical generators and loads, for example through load shedding, but the actual implementation has to be taken into account. DSO shedding capabilities have been designed to face safety issues that are caused by faults/errors in demand forecasting but not for cyber attacks. Therefore, mitigation strategies should be approached from two different perspectives: On one side, there is the need to strengthen cybersecurity countermeasures, including developing proper cybersecurity monitoring tools that adapt to IoT devices. On the other side, DSOs should maintain the capability to disconnect electrical generators independently from the operation of energy community platforms.

## 7. Conclusions

This paper analyzed the consequences of cyber attacks on grid-integrated RECs. Both low- and medium-voltage systems were subjected to power manipulation attacks while keeping in mind the restraints set for the installation of energy communities. The results also indicate how manipulating certain parameters of these power systems creates different plausible scenarios of grid vulnerability. The power manipulations led to either the injection or absorption of large amounts of power into the system, and both situations are dangerous. For these reasons, the risk to the distribution grids associated with the implementation of smart energy communities is high. While REC designers must implement proper cybersecurity countermeasures, such as proper cybersecurity monitoring tools, DSOs have to maintain the capability to disconnect generators and loads to mitigate the impact of attacks.

**Author Contributions:** Conceptualization, G.B.G.; Methodology, G.B.G.; Software, A.M.; Investigation, A.M.; Data curation, A.M.; Writing—original draft, A.M.; Writing—review & editing, G.B.G.; Supervision, M.M.; Project administration, M.M.; Funding acquisition, M.M. All authors have read and agreed to the published version of the manuscript.

**Funding:** This work was partially supported by project SERICS (PE00000014) under the MUR National Recovery and Resilience Plan funded by the European Union—NextGenerationEU: PE00000014.

**Institutional Review Board Statement:** Not applicable.

**Informed Consent Statement:** Not applicable.

**Data Availability Statement:** Data are contained within the article.

**Conflicts of Interest:** The authors declare no conflicts of interest.

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
