# Peer review of "Impact Analysis of Cyber Attacks against Energy Communities in Distribution Grids"

_electronics, doi:10.3390/electronics13091709_

Round 1
Reviewer 1 Report (Previous Reviewer 2)
Comments and Suggestions for Authors
The authors have addressed the various points and have greatly enhanced the paper. For example, Section 6 now makes some points on prospective mitigation. The Use Case Scenarios context has been reinforced. The introduction has been re-scoped, and the conclusion has been re-formulated.
Comments on the Quality of English LanguageFine.
Author Response
Dear reviewer, thank you very much for your suggestions and your positive feedback.
Reviewer 2 Report (New Reviewer)
Comments and Suggestions for Authors
The strength of this article is the impact analysis of cyber-attacks on energy communities in distribution grids. This topic is essential for the development of renewable energy, and it is excellent that this topic is handled in the journal of Electronics.
1. In row 48, the abbreviation DSO is used for the first time. Of course, it is known to specialists, but it needs explanation for more comprehensive readers.
2. The abbreviation MV is used for the first time in row 213. The comments are the same as in the previous section.
3. In row 217, the grid voltage level is 416 V. It is not understandable why such a voltage is used, and it needs explanation.
4. Fig. 5 needs explanation. In the upper part are numbers and words 901.0 GWh Actual load. This number probably represents the total electricity yearly demand in Italy. The load unit is GW, not GWh.
5. Table 1 names the terms Injection and Absorption. What do these terms mean? They are not explained in the text.
6. What means in Table 2 abbreviations (ind.) and (cap.)
7. What are load 1, load 2 and load 3 in figures 7, 8 and 9?
8. It is not clear why are the supplementary material.
Author Response
The strength of this article is the impact analysis of cyber-attacks on energy communities in distribution grids. This topic is essential for the development of renewable energy, and it is excellent that this topic is handled in the journal of Electronics.
Dear reviewer, thank you for your time and for your very precise suggestions. Below you can find the changes we made to improve the quality of the paper.
1. In row 48, the abbreviation DSO is used for the first time. Of course, it is known to specialists, but it needs explanation for more comprehensive readers.
2. The abbreviation MV is used for the first time in row 213. The comments are the same as in the previous section.
Both the abbreviations have been clarified when mentioned for the first time in the text.
3. In row 217, the grid voltage level is 416 V. It is not understandable why such a voltage is used, and it needs explanation.
We used the default parameters of the IEEE European Low-Voltage Test Feeder (ELVTF). We added a reference of the model in the corresponding line.
4. Fig. 5 needs explanation. In the upper part are numbers and words 901.0 GWh Actual load. This number probably represents the total electricity yearly demand in Italy. The load unit is GW, not Gwh.
The figure has been taken from the website of the Italian Transmission System Operator Terna s.p.a., as we clarified in the text. That number is the overall consumption of energy on the same day. Nevertheless, since that number was confusing in the Figure, we simply removed in. We explained the figure in the text, as follows:
“For example, a generic daily load demand trend of Italy is shown in Figure 5. The Figure shows the trend of instant active power demand in the whole country on a generic day of January (that is the coldest month of the year), taken from the website of the Italian Transmission System Operator Terna s.p.a. [32]; it can be noticed that the load demand rises from 8 am and reaches its peak around 11 am. It then starts to show a decline from 12 pm onwards and again reaches a peak in the
evening around 6 pm. The load demand is lower around 11 pm and keeps on declining.”
5. Table 1 names the terms Injection and Absorption. What do these terms mean? They are not explained in the text.
We clarified the meaning of the terms reported into the table in section 4.1 as follows:
“Within the same use case, the power manipulation attack is executed in three different scenarios, as shown in Table 1. The power manipulation attack can lead to higher absorption of power from the grid to the loads (in the case of controllable loads, such as Heat Ventilation and Air Conditioning systems) or to a higher production (injection) of power to the grid (for example, in case of Battery Energy Storage Systems). We simulated three different scenarios in which it ispossible to control a different magnitude of power across the three areas, considering a realistic deployment of resources across the distribution grid. For example, area 2 is subjected to higher magnitudes of power manipulations due to the centralized position of loads.”
6. What means in Table 2 abbreviations (ind.) and (cap.)
The abbreviations have been clarified in the text as follows:
“whereas the next two deal with the injection of inductive (".ind" in Table 2) and capacitive (".cap" in Table 2) reactive power. The study of reactive power at MV stage becomes relevant as high magnitudes of reactive power can also alter the grid profile.”
7. What are load 1, load 2 and load 3 in figures 7, 8 and 9?
We clarified the locations of Load 1, 34 and 50 in Section 5.1.1 as follows:
“he plot in this figure shows the voltage taken at load 1, load 34, and load 50, with respect to Figure 3 between 8 am and 11 am. The measure taken near load 1 corresponds to the voltage at the substation, while Loads 34 and 50, being located at two different distances from the substation, are representative of the voltage profile across the line..”
8. It is not clear why are the supplementary material.
While submitting the revised manuscript last time, we wanted to provide a version of the manuscript which consisted of the highlighted changes. That is why we added it in the supplementary material.

This manuscript is a resubmission of an earlier submission. The following is a list of the peer review reports and author responses from that submission.
Round 1
Reviewer 1 Report
Comments and Suggestions for Authors
The Introduction part should include the main aim of the paper, the authors' contribution, novelty, and the scientific problem to be addressed. Now the authors' contribution is presented as an aim, making it more difficult to define the novelty of this work.
The section for related works needs strong improvements. The authors should add more 10–20 references by providing not only what is presented in these works but also analyzing their limitations within the scope of this work. The table of the analyzed solutions can be added as well.
Self-citation is detected in Section 2. The authors analyze their own works [8, 9].
The attack model is not clearly described. It is recommended to add a diagram or concept of how the cyber attack can appear, where it can occur, and which parts it can target in the architecture of the energy community or energy communication.
It is not clear what is proposed in this paper. Any solution for the secure energy architecture implementation or review of threats to it? This work lacks a methodology section.
Minor editing of English language required
Author Response
Reviewer 1
Dear reviewer, thank you for your time and very precise suggestion. Below you can find the changes we made in the text to address your concerns.
“The Introduction part should include the main aim of the paper, the authors' contribution, novelty, and the scientific problem to be addressed. Now the authors' contribution is presented as an aim, making it more difficult to define the novelty of this work.
It is not clear what is proposed in this paper. Any solution for the secure energy architecture implementation or review of threats to it? This work lacks a methodology section.”
We modified the introduction of the paper to clarify the scope of the paper as follows:
“The main contribution of this paper is to analyze the risk associated with cyber attacks on smart Energy Communities, through the quantitative analysis of the impact on the distribution grid. Results can be used during a risk assessment phase of new energy communities, avoiding building custom simulations, thanks to the generalizable use cases of this work.
(…)
The results can be used by distribution system operators and RECs designers to minimize the risk during the development of the system; in particular, results show how, under certain conditions, the impact on the grid could be very high, therefore requiring more stringent security countermeasures.”
“The section for related works needs strong improvements. The authors should add more 10–20 references by providing not only what is presented in these works but also analyzing their limitations within the scope of this work. The table of the analyzed solutions can be added as well.”
11 references have been added. In particular, we integrated Section with the following paragraph:
Extensive initiatives have been undertaken within the power sector to address cybersecurity concerns related to smart grid. One significant effort is the work done by the National Electric Sector Cybersecurity Organization Resource (NESCOR)[1]. NESCOR has outlined the architecture and established cybersecurity requirements for Distributed Energy Resources (DERs) based on the framework proposed in the National Institute of Standards and Technology Interagency Report 7628[ 2]. The authors in [ 3] studied the consequences of a Denial of Service (DoS) attack that renders the active power dispatch message inaccessible from the Energy Management System (EMS) to the Energy Storage System (ESS). This attack becomes particularly detrimental when the power output of intermittent Distributed Energy Resources (DERs) such as wind turbines or photovoltaic (PV) systems abruptly declines due to changes in weather conditions, resulting in power imbalances and frequency deviations.
(…)
[ 12 ] discusses a scenario where Distributed Energy Resources (DERs) are compromised in a cyberattack, resulting in the manipulation of their power output. This could lead to continuous oscillations or even destabilize the system. However, the authors did not deeply explore how manipulating DERs in specific locations could amplify the impact of such attacks. Similarly, [ 13 ] assessed the ramifications of controlling a multitude of DERs, albeit with a focus on storage solutions. [14 ] demonstrated potential violations of voltage limits via cyberattacks on DERs within the context of the CIGRE medium voltage benchmark grid. Despite these insights, there remains a scarcity of research conducting simulations that account for the unique features and limitations of utilizing a single technology to manage a large array of DERs
“The attack model is not clearly described. It is recommended to add a diagram or concept of how the cyber attack can appear, where it can occur, and which parts it can target in the architecture of the energy community or energy communication. “
We clarified the attack model in section 3,3, also adding a Figure, as follows:
“The attack model scheme is shown in Figure 2. As shown, the scheme depicts the REC connection including the internet services and the power grid. The data being communicated by the REC via the smart gateway to the management system is measured by the sensors and smart meters such as the active and reactive power, frequency, voltage, current, and power factor. In turn, the management system handles the input and output of quantities by controlling the REC infrastructure. The attacker can thus, manipulate the data being used to control the REC devices in two main ways: from one side, he can exploit vulnerabilities of the EMS server, such as Cross-site Scripting [ 19 ], SQL Injection and Cross-site Request Forgery [20]. Also, he can target the smart gateways, that, since they are usually IoT devices built on low computational power devices, may suffer from severe vulnerabilities [21 ]. In both cases, the impact is the manipulation of the load parameters, active and reactive power references, and switching the operation mode of the energy-producing devices”
Reviewer 2 Report
Comments and Suggestions for Authors
Regarding the paper, “Impact Analysis of Cyber-attacks Against Energy Communities in Distribution Grids,” the approach vector (in addressing what is outlined in the abstract) misses a fundamental aspect of cyber – the counterpoising of the Confidentiality-Integrity-Availability (CIA) model. For this case, which involves Distribution Grids (DG), there is ongoing discourse regarding the potential inversion of the model; that is to say, that Availability-Integrity-Confidentiality has become an operative paradigm. The intricacies of handling that paradigm, particularly since renewables (the purported focus of the paper) introduces heightened instability into the system and therefore needs an enhanced monitoring paradigm, is not discussed within the paper. An adequate literature review is also not provided. Interestingly, statements, such as that stated in the Introduction, “the main hypothesis is that the central platform contains common vulnerabilities for web servers that could be exploited by attackers” runs somewhat counter to the paradigm, in many cases, wherein decentralized and microgrid strategies are adopted by various Renewable Energy Communities (RECs), which is a term referenced more than 25 times in the paper. Yet, there is no discussion on how DGs might handle the Common Vulnerabilities and Exposures (CVEs) provided on the MITRE CVE List or on the, say, commonly utilized National Institute of Standards and Technology (NIST) National Vulnerability Database (NVD), or even the Shodan platform for exposed Internet Protocol (IP) addresses of the DGs. There is no discussion on how currently available solutions might mitigate against some of the authors’ points. For example, on page 12 of 13, the paradigm of “islanding” is discussed (but the term is not used) as a problem, but it is not balanced against approach vectors for mitigation. This is a case, again, wherein some sections of the papers purport to be technical, but key aspects are not treated technically. In many cases, such as in Europe, where the authors are from, the DG is also operated the Transmission System Operator (TSO). In these cases, and others, the authors’ conclusion that “there are not many ways in which the grid is equipped to handle surplus amounts of power” is not necessarily true, as load shedding is a common form of offloading surplus power. The organization of the Use Case Scenarios (e.g., 4.1 and 4.2) as well as the further Case Studies (e.g., 5.1.1, 5.1.2, 5.1.3, etc) is confusing and reflects a shallow literature review. The Abstract and Introduction are too broad for what is, ultimately, introduced in the paper, and the Conclusion is not supported by the paper (and is not quite accurate). Section 3 on page 2 of 13 refers to "Cybersecurity Issues in Renewable Energy Communities," but the assertions in Section 3.2 and 3.3 are far too simplified (and not necessarily completely valid), and the paper is not convincing for what it set out to do.
Comments on the Quality of English LanguageQuality of English is fine.
Author Response
Reviewer 2
Dear reviewer, thank you for your time and very precise suggestion. Below you can find the changes we made in the text to address your concerns.
Interestingly, statements, such as that stated in the Introduction, “the main hypothesis is that the central platform contains common vulnerabilities for web servers that could be exploited by attackers” runs somewhat counter to the paradigm, in many cases, wherein decentralized and microgrid strategies are adopted by various Renewable Energy Communities (RECs), which is a term referenced more than 25 times in the paper.
We took into account the most common architecture employed for energy communities, both in scientific paper and commercial products. We strenghtened this assumption in the text, also by providing some references, as follows;
“ Notably, there are various commercial solutions available for such platforms. One such product is the Regalgrid platform [15] , as described by the manufacturer, it enables sophisticated management of energy resources by interfacing with various device types via the SNOCU controller [16 ]. This facilitates a comprehensive view of one’s energy profile and optimizes consumption management. The SNOCU controller exemplifies an advanced gateway that provides remote oversight of generators, often referred to as a "smart gateway" by its producers. Another innovation is the ROSE Energy Platform[17 ] a cloud-based service for establishing, simulating, and overseeing energy communities. This service includes an Energy Management System (EMS) module and a mobile application designed to engage community members. Additionally, ER-LIBRA CE [ 18 ] represents a cloud-based solution dedicated to the administration of Energy Communities and Self-Consumption Groups, incorporating a control module for storage system management. We notice how most of the currently available commercial products are based on a centralized platform, similar to the reference architecture shown in Figure 1”
Regarding the paper, “Impact Analysis of Cyber-attacks Against Energy Communities in Distribution Grids,” the approach vector (in addressing what is outlined in the abstract) misses a fundamental aspect of cyber – the counterpoising of the Confidentiality-Integrity-Availability (CIA) model. For this case, which involves Distribution Grids (DG), there is ongoing discourse regarding the potential inversion of the model; that is to say, that Availability-Integrity-Confidentiality has become an operative paradigm. The intricacies of handling that paradigm, particularly since renewables (the purported focus of the paper) introduces heightened instability into the system and therefore needs an enhanced monitoring paradigm, is not discussed within the paper. An adequate literature review is also not provided.
Yet, there is no discussion on how DGs might handle the Common Vulnerabilities and Exposures (CVEs) provided on the MITRE CVE List or on the, say, commonly utilized National Institute of Standards and Technology (NIST) National Vulnerability Database (NVD), or even the Shodan platform for exposed Internet Protocol (IP) addresses of the DGs.
We discussed the role of the MITRE CVE List in Section 6:
While, from one side, several services can be used to monitor known vulnerabilities on the employed devices, such as the Common Vulnerabilities and Exposures (CVEs) provided on the MITRE CVE List, it is practically unfeasible to dispose of an updated asset inventory for the huge variety and quantity of devices managed by a single utility. Moreover, cybersecurity in the Operational Technology environment must take into account attack response strategies that guarantee the availability of the main power system.
There is no discussion on how currently available solutions might mitigate against some of the authors’ points. For example, on page 12 of 13, the paradigm of “islanding” is discussed (but the term is not used) as a problem, but it is not balanced against approach vectors for mitigation. This is a case, again, wherein some sections of the papers purport to be technical, but key aspects are not treated technically. In many cases, such as in Europe, where the authors are from, the DG is also operated the Transmission System Operator (TSO). In these cases, and others, the authors’ conclusion that “there are not many ways in which the grid is equipped to handle surplus amounts of power” is not necessarily true, as load shedding is a common form of offloading surplus power.
A discussion on mitigation strategies has been added in Section 6:
“This risk has to be taken into account, especially since little automation is generally implemented in low-voltage grids. DSO can generally decide to disconnect electrical generators and loads, for example through load shedding, but the actual implementation has to be taken into account. DSO shedding capabilities have been designed to face safety issues, caused by faults/errors in demand forecasting, but not for cyber attacks. Therefore, mitigation strategies should be approached from two different perspectives: from one side, there is the need to strengthen cybersecurity countermeasures, also by developing proper cybersecurity monitoring tools that adapt to IoT devices. On the other, DSO should maintain the capability to disconnect electrical generators independently from the operation of energy community platforms.”
The organization of the Use Case Scenarios (e.g., 4.1 and 4.2) as well as the further Case Studies (e.g., 5.1.1, 5.1.2, 5.1.3, etc) is confusing and reflects a shallow literature review.
We clarified the structure of the use case scenarios in section 4 as follows:
“Initially, the power constraints and boundaries on RECs were defined by Article 42-bis of Law No. 8 [ 19 ] allowing them to be only located on low-voltage grids and restricting them to generate and consume plant power not more than 200kW. It was later updated by Legislative Decree No. 199 [20 ] which eased the above power constraints from 200kW to 1MW allowing a larger number of prosumers to participate. However, taking into account general examples with low-voltage distribution systems such as residential buildings [21 ], a deployment of 200kW is considered to be optimal. We take into account the two main cases that, according to national laws that establish economic incentives for RECs, could be designed by engineers: a maximum of 200kW under a single LV substation, and a maximum of 1 MW under a single primary MV substation.”
The Abstract and Introduction are too broad for what is, ultimately, introduced in the paper, and the Conclusion is not supported by the paper (and is not quite accurate).
We modified the introduction of the paper to clarify the scope of the paper as follows:
“The main contribution of this paper is to analyze the risk associated with cyber attacks on smart Energy Communities, through the quantitative analysis of the impact on the distribution grid. Results can be used during a risk assessment phase of new energy communities, avoiding building custom simulations, thanks to the generalizable use cases of this work.
(…)
The results can be used by distribution system operators and RECs designers to minimize the risk during the development of the system; in particular, results show how, under certain conditions, the impact on the grid could be very high, therefore requiring more stringent security countermeasures.”
The conclusions have been re-formulated as follows:
“For these reasons, the risk for the distribution grid associated with the implementation of smart energy communities is high. While REC designers must implement proper cybersecurity countermeasures, such as proper cybersecurity monitoring tools, DSOs have to maintain the capability to disconnect generators and loads to mitigate the impact of attacks.”
Section 3 on page 2 of 13 refers to "Cybersecurity Issues in Renewable Energy Communities," but the assertions in Section 3.2 and 3.3 are far too simplified (and not necessarily completely valid), and the paper is not convincing for what it set out to do.
We took into account the most common architecture employed for energy communities, both in scientific paper and commercial products. We strenghtened this assumption in the text, also by providing some references, as follows;
We also deepened the attack model, also adding a Figure, as follows:
“The attack model scheme is shown in Figure 2. As shown, the scheme depicts the REC connection including the internet services and the power grid. The data being communicated by the REC via the smart gateway to the management system is measured by the sensors and smart meters such as the active and reactive power, frequency, voltage, current, and power factor. In turn, the management system handles the input and output of quantities by controlling the REC infrastructure. The attacker can thus, manipulate the data being used to control the REC devices in two main ways: from one side, he can exploit vulnerabilities of the EMS server, such as Cross-site Scripting [ 19 ], SQL Injection and Cross-site Request Forgery [20]. Also, he can target the smart gateways, that, since they are usually IoT devices built on low computational power devices, may suffer from severe vulnerabilities [21 ]. In both cases, the impact is the manipulation of the load parameters, active and reactive power references, and switching the operation mode of the energy-prod